# Isolation and Characterization of the *GmMT-II* Gene and Its Role in Response to High Temperature and Humidity Stress in *Glycine max*

**DOI:** 10.3390/plants11111503

**Published:** 2022-06-03

**Authors:** Sushuang Liu, Yanmin Liu, Chundong Liu, Yang Li, Feixue Zhang, Hao Ma

**Affiliations:** 1Department of Life Sciences and Health, Huzhou College, Huzhou 313000, China; liu@zjhzu.edu.cn (S.L.); liuchundong@zjhzu.edu.cn (C.L.); 2College of Life Science, Huzhou University, Huzhou 313000, China; 02572@zjhu.edu.cn; 3Institute of Crop, Huzhou Academy of Agricultural Sciences, Huzhou 313000, China; feixue66666@126.com; 4State Key Laboratory of Crop Genetics and Germplasm Enhancement, Nanjing Agricultural University, Nanjing 210095, China

**Keywords:** antioxidants, *GmMT-II*, seed protein, soybean, seed quality

## Abstract

Metallothioneins (MTs) are polypeptide-encoded genes involved in plant growth, development, seed formation, and diverse stress response. High temperature and humidity stress (HTH) reduce seed development and maturity of the field-grown soybean, which also leads to seed pre-harvest deterioration. However, the function of MTs in higher plants is still largely unknown. Herein, we isolated and characterized the soybean metallothionein II gene. The full-length fragment is 255 bp and encodes 85 amino acids and contains the HD domain and the *N*-terminal non-conservative region. The subcellular location of the GmMT-II-GFP fusion protein was clearly located in the nucleus, cytoplasm, and cell membrane. The highest expression of the *GmMT-II* gene was observed in seeds both of the soybean Xiangdou No. 3 and Ningzhen No. 1 cultivars, as compared to other plant tissues. Similarly, gene expression was higher 45 days after flowering followed by 30, 40, and 35 days. Furthermore, the *GmMT-II* transcript levels were significantly higher at 96 and 12 h in the cultivars Xiangdou No. 3 and Ningzhen No. 1 under HTH stress, respectively. In addition, it was found that when the Gm1-MMP protein was deleted, the *GmMT-II* could bind to the propeptide region of the Gm1-MMP, but not to the signal peptide region or the catalytic region. *GmMT-II* overexpression in transgenic *Arabidopsis* increased seed germination and germination rate under HTH conditions, conferring enhanced resistance to HTH stress. *GmMT-II* overexpressing plants suffered less oxidative damage under HTH stress, as reflected by lower MDA and H_2_O_2_ content and ROS production than WT plants. In addition, the activity of antioxidant enzymes namely SOD, CAT, and POD was significantly higher in all transgenic *Arabidopsis* lines under HTH stress compared wild-tpye plants. Our results suggested that *GmMT-II* is related to growth and development and confers enhanced HTH stress tolerance in plants by reduction of oxidative molecules through activation of antioxidant activities. These findings will be helpful for us in further understanding of the biological functions of MT-II in plants.

## 1. Introduction

Soybean (*Glycine max* [L.] Merr.) is one of the most important leguminous crops widely cultivated for protein and edible oil sources [1]. Soybean seed contains 18–25% oil and 40–44% crude protein [2]. Its production has been steadily increased in the combination of plant breeding and genetic improvement techniques. However, environmental stresses such as salinity, drought, extreme temperature, high humidity, waterlogging, metal toxicity, pollutants, ultraviolet radiation, and other factors drastically reduce soybean yield [3,4,5,6] and these stresses alter plant physiological and biochemical activities, including stomatal conductance, electron transport, CO_2_ diffusion, carboxylation efficiency, water use efficiency, respiration, transpiration, photosynthesis, and membrane functions [3]. Furthermore, environmental stresses adversely affect soybean production along with reduced yield quality in terms of lower protein and oil content in seed [7]. Among the abiotic stresses, high temperature and drought are the important stressors that can affect soybean seed yield and quality [8]. High-temperature stress reduced isoflavone concentration and associated gene expression in both pods and seeds of soybean [9]. It was found that high-temperature and humidity (HTH) stress during the physiological maturity period (R7 period) of soybean seed is a significant factor leading to seed deterioration prior to harvest [10]. Nakagawa et al. [8] reported that high temperature alters lipids and protein concentration in soybean seeds compared to low temperature. A higher number of pods was observed in soybean plants under elevated temperature (38/30 °C day/night) [11]. Furthermore, relative humidity (RH) is also one of the vital abiotic factors, higher RH directly affects plant growth and development by disturbing the transpiration and nutrient uptake from soil or growing medium [12]. Many researchers have reported that high temperature and humidity (HTH) leads to pre-harvest seed deterioration such as abnormal seed production, germination rate reduction, seed vigor, storage ability, and seed quality of soybean [13,14,15]. In our previous comparative transcriptomic and proteomic studies, we revealed that HTH stress in the development of soybean seeds changed the expression of associative genes and function of proteins. Shu et al. [14] identified 1081 and 357 differentially expressed genes (DEGs) in Ningzhen No. 1 and Xiangdou No. 3, respectively under HTH stress and these are related to photosynthesis, carbohydrate metabolism, lipid metabolism, and heat shock proteins. Furthermore, a total of 42 proteins were identified and matched with 31 diverse protein species, which are involved in various metabolic and cellular functions in pre-harvest seed deterioration of soybean under HTH conditions [15]. However, the molecular mechanism of soybean seed response to HTH stress is still largely unknown.

Metallothioneins (MTs) are gene-encoded polypeptides, constitute an extremely diverse family of ubiquitous, low-molecular weight (60 amino acids, 4–8 kDa), cysteine-rich (30%) proteins that bind to metals in a variety of organisms ranging from yeast to vertebrates and plants. MT proteins are classified into three sub-classes: MT-I, MT-II, and MT-III [16]. Based on their amino acid sequences, plant class MT-II proteins are classified into MT1, MT2, MT3, and MT4. In plants, MT-II responds to a variety of environmental stresses, such as heavy metals, drought, salinity, salt and alkali conditions [17,18,19]. Several MT genes are identified and expressed in various tissues of plants, such as leaves, stems, ripening fruits, and wounded tissues [20]. Studies have reported that the *MT2* gene reduced cell oxidative damage by scavenging reactive oxygen species and enhancing the metal chaperon [18,19,20]. Both temperature and humidity stresses often occur simultaneously, but very few studies have been reported about their combined effect on plants. HTH is a key environmental factor in southern China and the United States, which significantly reduces soybean seed quality through pre-harvest seed deterioration [14]. The molecular mechanism of stress tolerance is a complex process, and to understand this deeply, it requires information at the genomics, transcriptomics, and proteomics level [4]. A previous study showed that *GmSBH1* is involved in response to HTH stress in developing soybean seeds [13]. However, the molecular mechanism of soybean response to high temperature and humidity and its effective management is still largely unknown. In this study, we isolated and evaluated the *MT-II* gene from pre-harvest seed deterioration-sensitive (Ningzhen No. 1) and resistant (Xiangdou No. 3) soybean cultivars and also investigated the significance of this gene in soybean plant response to HTH stress. Our findings revealed that *GmMT-II* showed diverse expression patterns among the various tissues and was involved in enhancing tolerance to HTH stress. Overexpression of *GmMT-II* could mitigate the oxidative damage in *Arabidopsis* exposed to HTH stress. These will lead to better understanding of the biological functions of the Gm*MT-II* in growth and development of plants, including the response to abiotic stresses. 

## 2. Results

### 2.1. Sequence Analysis of GmMT-II in Glycine Max

Using the NCBI-published soybean GmMT-II CDS sequences as the template, the above cDNA was amplified by PCR using appropriate primers and we found that there was a clear specific band at the 379 bp position (Figure 1A). The open reading frame (ORF) of *GmMT-II* was obtained from the cDNA in *G. max*. The full-length fragment contains 255 bp and encodes 85 amino acids (GenBank accession number: NM_001250480). The BLAST results showed that GmMT-II was homologous with the protein sequences of different plant MTs. The phylogenetic tree showed that the GmMT-II protein belongs to the fourth subgroup of MTs and the GmMT-II protein sequences had the highest similarity with different plant MTs such as OsMT4 from *Oryza sativa*, AtMT4A from *Arabidopsis thaliana*, AhMT4 from *Arachis hypogaea*, and TaMT4 from *Triticum aestivum* (Figure 1B). 

Furthermore, we aligned the amino acid sequence encoded by the GmMT-II gene with Type 4 MT proteins of other species, including Ah: *Arachis hypogaea*, At: *Arabidopsis thaliana*, Os: *Oryza sativa*, Ta: *Triticum aestivum*, and the results are shown in Figure 1B. The sequence identity of the product to its homologous protein was 54.71%. It was further found that all type 4 MTs have three conserved Cys-rich domains, each domain has 5–6 Cys residues, the domains are spaced by 10–16 amino acid residues, and Cys mostly exists in the structure of Cys-Xaa-Cys (Figure 1C).

### 2.2. Expression of the GmMT-II Gene in Glycine max

The transcript level of *GmMT-II* was analyzed in Xiangdou No. 3 and Ningzhen No. 1 soybean cultivars in different time intervals under HTH stress conditions (Figure 2A,B). The results showed that *GmMT-II* expression was significantly higher in Xiangdou No. 3 cultivar at 96 h followed by 168 h and 48 h, whereas no significant differences were observed between control and stress treatment at 0, 12, and 24 h, even though the gene expression was higher in HTH conditions. Similarly, *GmMT-II* was significantly higher in Ningzhen No. 1 cultivar at 12 h. The gene expression was also higher in Ningzhen No. 1 cultivar under HTH conditions at 0, 24, 96, and 168 h compared to the control plants, but no significant differences were observed between the treatments (Figure 2A,B).

On the other hand, the gene expression of *GmMT-II* was also investigated in Xiangdou No. 3 and Ningzhen No. 1 soybean cultivars after flowering of the plants. The *GmMT-II* gene expression was significantly higher in the Xiangdou No. 3 cultivar 40 and 45 days after flowering compared to the Ningzhen No. 1 cultivar (Figure 2C). However, the *GmMT-II* gene expression was significantly higher in the Ningzhen No. 1 cultivar after 30 and 35 days following the flowering compared to Xiangdou No. 3 cultivar. Similarly, the expression of *GmMT-II* was observed in root, stem, leaf (young, mature, and old), flower, pod and seeds of Xiangdou No. 3 and Ningzhen No. 1 soybean cultivars. The results showed that *GmMT-II* gene expression was significantly higher in seeds both of the soybean cultivars (Figure 2D). A significant increase of *GmMT-II* gene expression was observed in pods of Xiangdou No. 3 cultivar compared with that of Ningzhen No. 1 cultivar. Furthermore, lower gene expression was observed in root, stem, leaf (young, mature, and old), and flower of soybean cultivars and no significant differences were observed among them (Figure 2D).

### 2.3. Subcellular Localization of GmMT-II Protein

In the current experiment, the constructed fusion expression vectors were infiltrated into onion epidermal cells by the particle bombardment method, and the green fluorescence was observed by a laser confocal microscope. The GmMT-II-GFP fusion protein was similar to the control pA7-GFP protein without the GmMT-II gene, and both were clearly distributed in the whole cell, indicating that the GmMT-II gene was located in the nucleus, cytoplasm, and cell membrane (Figure 3).

### 2.4. Screening of Gm1-MMP and GmMT-II interaction sites in Yeast

The GmMT-II protein has three Cys-rich domains, and we divided them into full-length (1–85 aa), *N*-terminal (1–34 aa), and *C*-terminal (35–85 aa) domain. Furthermore, based on the main conserved domains of Gm1-MMP proteins, such as Signal peptide (SP), Propeptide (PP), Catalytic domain (CD), we classified Gm1-MMP proteins into four domains, namely full-length (1–305 aa), *S*-terminal (1–16aa), *P*-terminal (17–135 aa), and *C*-terminal (136–305 aa). To better understand the binding site between the Gm1-MMP primer protein and GmMT-II trap protein, different deletion vectors were constructed based on the characteristics of the conserved domains of the GmMT-II protein and Gm1-MMP protein as well as intercellular co-transformation into yeast cells NMY32. The binding form between Gm1-MMP and GmMT-II was evaluated by screening the SD-TL medium and SD-THLA medium, and X-α-gal staining. The experimental results showed that both the *N*-terminal and *C*-terminal of GmMT-II can bind to the Gm1-MMP protein, indicating that it has a Cys-rich domain, and it can bind to the Gm1-MMP protein (Figure 4). It was found that when the Gm1-MMP protein was deleted, the pro-peptide region of the Gm1-MMP protein could bind to the GmMT-II but not the signal peptide region or the catalytic region.

### 2.5. Response of the GmMT-II Overexpression Transgenic Arabidopsis to HTH Stress

To evaluate the effect of high temperature and humidity (HTH) on the seed germination rate, three overexpression transgenic *Arabidopsis* lines ((L2, L3, L4, and WT) were used and seeds are seeded in an MS medium [21] followed by exposure to HTH (Figure 5). The results indicated that seed germination of wild-type and overexpression transgenic lines of *Arabidopsis* started after 24 h of incubation in both control and HTH stress conditions. The germination rate of WT and all overexpression transgenic lines were nearly similar under control conditions, but their germination rate varied under HTH treatment. The highest seed germination rate was observed in the L2 compared to L3 and L4 overexpression transgenic *Arabidopsis* under HTH conditions. The lowest seed germination rate was observed in wild-type *Arabidopsis* under HTH conditions. These results suggest that overexpressing plants are more tolerant to HTH stress than wild-type plants. 

### 2.6. GmMT-II Overexpression Transgenic Arabidopsis Lines Reduces Oxidative Stress under HTH

To further verify the HTH stress tolerance phenotype, we measured several physiological indicators in plant leaves, including H_2_O_2_ content and ROS production content, as well as stress marker components such as malondialdehyde (MDA). The results showed that the most oxidative stress markers namely H_2_O_2_ content and ROS production rate in leaves of WT plants and transgenic plants were significantly increased under HTH stress as compared with those of normal plants, but these stress markers in overexpressed plants leaves had less increased than in WT plants (Figure 6). The microscopic observation results also indicated a higher brown color precipitation observed in the wild type compared to L4 and L3 transgenic *Arabidopsis* under HTH conditions. No obvious brown spots were observed in the control plants of wild-type and overexpression transgenic *Arabidopsis* plants, indicating that less oxidative damage occurred in the overexpressed *GmMT-II* plants (Figure 6A). The higher accumulation of H_2_O_2_ was observed in the L3, L4 transgenic lines and wild-type *Arabidopsis* under HTH conditions compared to the control plants, but the level of H_2_O_2_ in the wild type was significantly (*p* < 0.01) higher than that in the overexpression transgenic *Arabidopsis* (Figure 6B). On the other hand, ROS accumulation was also significantly (*p* < 0.01) higher in wild-type *Arabidopsis* plants compared to transgenic lines. The lower level of ROS accumulation was observed in the L3 transgenic lines, whereas ROS accumulation was significantly higher in L4 plants (Figure 6C).

### 2.7. GmMT-II Overexpression Transgenic Arabidopsis Lines Increase Antioxidant Activities under HTH Stress

In terms of antioxidant enzyme activity under control conditions, there was no significant difference between the four genotypes (Figure 7). In particular, SOD activity was significantly (*p* < 0.01) higher in all overexpression transgenic *Arabidopsis* lines (L2, L3 and L4) compared to WT plants under HTH conditions and the highest activity was observed in L2, while the lowest activity was observed in wild-type plants (Figure 7A). After HTH treatment, CAT activity was substantially increased both in WT and overexpressed lines and, of course, the CAT activity was highest in transgenic plants. The highest CAT activity among the four genotypes was observed in L3, and, certainly, WT showed the lowest activity under HTH condition (Figure 7B). Similarly, as shown in Figure 7C, POD activity increased substantially in WT and transgenic plants with a wider increasing range in transgenic lines, and the range increased in an order from L2, L4 to L3, resulting in the biggest activity in L2 lines, but no significant changes were observed in POD activity between control and HTH-treated wild-type *Arabidopsis* (Figure 7C). There were no significant differences in the the MDA content in leaf among the four plants tested under non-stressed condition, but it was noteworthy that the MDA content was slightly higher in WT plants. The MDA content was raised dramatically in the leaves of all tested plants exposed to HTH stress, whereas the MDA content *GmMT-II* overexpressing plants were less than that of WT plants, suggesting that *GmMT-II* overexpressing plants suffered less under HTH stress (Figure 7D). These results indicated that the enhanced antioxidant capacity in *GmMT-II* overexpression transgenic *Arabidopsis* plants helped alleviate oxidative damage caused by HTH stress, contributing to improving tolerance to HTH stress and plant growth. 

## 3. Discussion

In the natural environment, plants are frequently subjected to different types of environmental stresses and, therefore, they have to adapt with multiple stresses simultaneously. High temperature and humidity are the major abiotic factors that reduced root, shoot, seed yield, seed germination and seed quality, and plant physiological parameters in soybean. Further molecular studies can help us to understand the internal mechanism of HTH stress in soybean. To elucidate this, we cloned and characterized *GmMT-II* gene from pre-harvest seed deterioration-sensitive (Ningzhen No. 1) and resistant (Xiangdou No. 3) soybean cultivars. Multiple sequences alignment with another plants MT-II and a phylogenetic evolution analysis confirmed that the domain of GmMT-II was highly conserved with type 4 of the class II MT family and showed 85% similarity (Figure 1). The MT superfamily combines a large variety of small cysteine-rich proteins and the maximum number of Cys residues was noticed in the MT4 protein followed by the MT2, MT1, and MT3 proteins containing 17, 14, 12, and 10 Cys residues, respectively [22]. Furthermore, the expression of MT genes varied in different tissues of plants [23]. In this study, the expression of the MT-II gene was significantly higher in the seeds followed by mature and old leaves of both soybean cultivars. Whereas *GmMT-II* was significantly higher in pods of Xiangdou No. 3 cultivar compared to Ningzhen No. 1 (Figure 2). Jin et al. [17] reported that *SsMT2* expression was significantly higher in *Arabidopsis* leaves and seeds, and its expression level increased when *S. salsa* plants were exposed to metal and salt stresses. Yuan et al. [24] observed high expression of *M**etallothionein2b* (*OsMT2b*) in *Oryza sativa,* germinating embryos, and primordium of lateral roots. Guo et al. [25] reported that MT genes are highly expressed in trichomes, during senescence, young leaves, and at root tips induced by Cu. Furthermore, the authors observed that the expression level of the genes MT1a and MT2b was significantly higher in the phloem of all organs; on the contrary, the expression of MT2a and MT3 was higher in mesophyll cells, while the expression of *MT4* genes was restricted to seeds. All of these results indicate that *GmMT-II* is related to the development of soybean. In addition, the higher *GmMT-II* expression was observed both in Xiangdou No. 3 and Ningzhen No. 1 cultivars under HTH conditions. Whereas lower *GmMT-II* expression was observed in the WT plants under same treatment conditions. Similar results were also found in other studies and they notified that the expression was significantly higher in the *Suaeda salsa* under metal and salt [17], Apium graveolens under metal, and various environmental stresses [26].

The *GmMT-II* gene CDSs in Xiangdou No. 3 and Ningzhen No. 1 soybean were abso- lutely identical. We found that there are differences in the expression of *GmMT-II* in the two varieties by analyzing the transcription level of *GmMT-II* in different tissues, seed development process, and under high temperature and high humidity stress. In order to study the reason, we isolated the promoter sequences of the *GmMT-II* gene in the two varieties which were found to be consistent. So, we speculate that the expression of the gene at the transcription level is very complex and may be affected by many pathways. In addition, this gene may also regulate many traits, namely it has the function of one cause and multiple effects. In multiple studies of our research group, we found that the CDS sequences of different genes are consistent, but there are differences in the expression between the two varieties.

In addition, although the expressions of *GmMT-**II* were different in the two varieties, they both had high expression level in seeds, indicating that it is involved in the development of seeds, and for the response to high temperature and high humidity stress. We pay more attention to the difference between the stress treatment and the control. The results showed that *GmMT-II* was significantly different between the stress treatment and the control, indicating that it was involved in the response to high temperature and high humidity stress. As for why the *GmMT-II* gene exists at the transcription level between different varieties, we will further study in the future, such as further study from DNA methylation and so on.

Higher germination reflects better growth and development of seedlings and thus leads to higher crop production [27]. Overexpression of the *SsMT2* gene in transgenic yeast improved the tolerance of the cells to NaCl, CdCl_2_ and H_2_O_2_[17]. Furthermore, better seed germination rates in *SsMT2* transgenic *Arabidopsis* lines under the same stress conditions compared to wild-type plants were obtained. The other study concluded that the expression of *SbMT2* was significantly higher in transgenic tobacco, helping to increase the seed germination rate under salt tolerance (NaCl), osmotic (PEG), and metal (Zn^++^, Cu^++^ and Cd ^++^) tolerance than wild-type tobacco plants [28]. In the present study, we observed that transgenic *Arabidopsis* had higher seed germination rates compared to wild-type *Arabidopsis* under HTH stress conditions (Figure 5). The overexpression of the *GmMT-II* gene in transgenic *Arabidopsis* improved the tolerance of developing seeds under HTH stress conditions.

In this study, *GmMT-II* transgenic *Arabidopsis* plants reduced H_2_O_2_ and ROS accumulation compared with wild-type *Arabidopsis* plants under HTH stress conditions (Figure 6). This observation was consistent with the results of MTs in other plant species. For example, Jin et al. [17] reported that the *SsMT2* gene in transgenic *Arabidopsis* lines increased H_2_O_2_ scavenging, which resulted in a lower accumulation of H_2_O_2_ in the transgenic plants under metal and salt stress conditions. Similarly, less H_2_O_2_ accumulation was observed in *GhMT3a* transgenic tobacco plants than in wild-type plants under such stress conditions [29]. Enzymatic (SOD, POD, CAT, and APX) and non-enzymatic (ascorbate, glutathione, and phenolic compounds) antioxidants are important ROS scavenging enzymes activated under different stress (high salinity, drought, and low temperature) conditions to maintain the ROS homeostasis [30]. Here, we observed that a significant level of SOD, CAT, and POD content in transgenic *Arabidopsis* lines under HTH stress conditions indicates the possible mechanism of *GmMT-II* in ROS scavenging/detoxification mechanism in transgenic *Arabidopsis* plants under stress conditions (Figure 7). Likewise, SOD, POD, and APX activities were significantly higher in *SbMT-2* transgenic tobacco plants compared to wild-type tobacco, which increases ROS detoxification and scavenging under salt, osmatic, and metal stress conditions [28]. The above results indicated that GmMT-II may be involved in inhibiting HTH-induced damage by regulating the antioxidant defense system and reducing ROS production.

## 4. Materials and Methods

### 4.1. Sequence Analysis of Metallothioneins in Glycine Max

*GmMT-II* was isolated using the gene-specific primers pair ORF-GmMT-II-F and ORF-GmMT-II-R (Appendix A). The PCR conditions were as follows: at 95 ℃ for 5 min and 30 cycles of 94 ℃ for 45 s, 56 ℃ for 60 s, and 72 ℃ for 120 s with a final extension at 72 ℃ for 10 min. Homology search was performed using blastn at NCBI (http://www.ncbi.nlm.nih.gov/blast/, accessed on 25 April 2021). The 18 MT proteins used for the phylogenetic tree are the following: CarMT2 (*Cicer arietinum*, CAA65009.1), PsMT2 (*Pisum sativum*, BAD18383.1), GmMT2 *(Glycine max*, NP_001235506.1), AtMT2A (*Arabidopsis thaliana*, NP_187550.1), AtMT2B (*Arabidopsis thaliana*, AAO42816.1), GmMT1 (*Glycine max*, BAD18376.1), CarMT1 (*Cicer arietinum*, CAA65008.1), MsMT1 (*Medicago sativa*, AAF04584.1), PsMT1 (*Pisum sativum*, BAD18383.1), OsMT4 (*Oryza sativa*, AAS78805.1), TaMT4 (*Triticum aestivum*, P30570.2), AtMT4A (*Arabidopsis thaliana*, NP_181731.1), GmMT4 (*Glycine max*, NP_001237409.1), AhMT4A (*Arachis hypogaea*, ABG57067.1), AtMT3 (*Arabidopsis thaliana*, AEE75656.1), TcMT3 (*Theobroma cacao*, EOY13005.1), OcMT3 (*Oryza coarctata*, ABV58318.1), ZmMT3 (*Zea mays*, NP_001105499.1). Phylogenetic tree analysis was performed by MEGA 6 [31] with the Neighbor-Joining method and bootstrap analysis (1000 replicates). Sequences alignment from various species were aligned using Clustal W (version 1.81, accessed on 25 April 2021) and Gene Doc (version 2.7, accessed on 25 April 2021) software.

### 4.2. Plant Materials and Stress Treatment

Two soybean cultivars (pre-harvest seed deterioration-sensitive cv. Ningzhen No. 1 and resistant cv. Xiangdou No. 3) were used in this study [15]. Seeds were sown in plastic pots and normally managed (30 °C/20 °C, 70% RH, and 10 h/14 h (light/dark)). For the tissue expression assay, roots and stems were collected at V1 stage; leaves and flowers were collected at R2 stage; pods were collected at R4 stage, and harvest mature seeds were collected at R8 stage. HTH stress on potted soybean plants (R 7 period) was performed according to Wang et al. [15]. Treated plants were transferred to a growth cabinet at 40 °C/30 °C, 100%/70% humidity (RH) and 10 h/14 h (light/dark) light cycle for 7 d. Parallelly, control plants in the equal growth stage were cultivated at normal conditions. Seeds were collected from the central part of 10 control plants and 10 treated plants after stress 0, 12, 24, 48, 96, and 168 h; each sample contains three independent biological replicates. All samples were immediately frozen in liquid nitrogen and stored at −80 °C.

Both the overexpression transgenic *Arabidopsis* lines (L2, L3, and L4) and wild-type *Arabidopsis* seeds were surface-sterilized and sown on plates containing 1/2 MS medium and transferred to growth chambers maintained at 40 °C temperature and 100% RH with a 12/12 h photoperiod as stressed treatment. Whereas, for control treatment, we maintained optimum growing conditions (28 °C and 70% RH with a 16/8 h photoperiod). Germination rate was evaluated as the percentage of germinated seeds against all sowed seeds of each genotype according to the method described by Brun et al. [32]. 

### 4.3. RNA Extraction, cDNA Library and qRT-PCR 

Total RNA was extracted from the ground leaves according to the manufacturer’s instructions (EASY spin RNA Plant Mini Kit, Aidlab, China). Then, 1μg of total RNA was reverse-transcribed into cDNA with HiScript^®^ II Q RT SuperMix for qPCR (Vazyme, China) following the instructions. The primer sequences were designed based on the corresponding gene sequence by searching the NCBI database (https://www.ncbi.nlm.nih.gov/tools/primer-blast/, accessed on 20 December 2020) and Beacon Designer 7.9 (Premier Biosoft International, CA, USA) was used to design the primers which were listed in Appendix A.

Gene expression analysis was performed using a qRT-PCR CFX96 thermocycler (Bio-Rad, accessed on 24 September 2021). The primers are listed in Appendix A (qRT-GmMT-II-F and qRT-GmMT-II-R), and the soybean Actin gene (accession No. V00450) was used as a standard control in RT-PCR reactions. The following thermal cycle conditions were used: 95 °C for 30 s, followed by 40 cycles of 95 °C for 5 s, 58 °C for 20 s, and 72 °C for 20 s. All reactions were performed in triplicate. Following the PCR, a melting curve analysis was performed. Ct or threshold cycle was used for relative quantification of the input target number. The relative gene expression levels were calculated according to the 2^–ΔΔCt^ method [33].

### 4.4. Subcellular Localization and Microscopic Analysis 

The full length CDS of *GmMT-II* was cloned into the binary vectors pA7-GFP using the ClonExpress TM II One Step Cloning Kit (Vazyme, Nanjing, China) and inserted into pA7-GFP vector to construct a GmMT-II-pA7-GFP fusion expression vector. The fusion construct and the empty control vector (pA7-GFP) were transformed into the *A.*
*tumefaciens* and transient transformation of onion epidermal was performed through the particle bombardment method. The plants were kept at 28 °C and 80% relative humidity (RH) with a 16/8 h photoperiod for 48 h until examination. Fluorescent signals were observed using a confocal laser scanning microscopy (CLSM) (Leica TCS SP2 confocal microscope, Jena, Germany). All CLSM images were obtained using Leica confocal software and the HCX PL APO 63×/1.2 W CORR water immersion objective. The GFP channel was acquired by excitation at 488 nm with detection at 500–530 nm. At least three replicates were used to essay all transient expression.

### 4.5. Screening of Gm1-MMP and GmMT-II Interaction Sites in Yeast

Homologous recombination was used to design specific primers for full-length PCR amplification (Appendix A). The full-length coding sequences (CDSs), S domain (1–16aa) P domain (17–135 aa) and C domain (136–305 aa) of the Gm1-MMP were cloned into pBT3SUC vector (Clontech, Shanghai, China) according to the different domain functions, respectively. Additionally, the full-length coding sequences (CDSs), N domain (1–34aa) and C domain (35–85 aa) of the GmMT-II were cloned into pPR3N vector (Clontech, Shanghai, China) (Appendix A). The constructed positive plasmids were used for the co-transformation of into yeast strain AH109 containing the His3 and LacZ reporter genes (Clontech, Shanghai, China), after which the protein–protein interactions were examined in selective medium lacking Trp, Leu, His, and Ade and supplemented with the optimal x-α-gal (Coolaber, Beijing, China).

### 4.6. Plasmid Construction and Transformation of Arabidopsis

To further investigate the function of GmMT-II, *GmMT-II*-overexpressing *Arabidopsis* transgenic lines were generated. The GmMT-II coding sequence was amplified with the GmMT-II-forward/reverse primer pair harboring the BamH I and Sac I sites (Appendix A). The PCR products were directly cloned into the pCAMBIA1301 cassette, containing the CaMV 35S promoter, to generate the plasmid 35S:GmMT-II. This plasmid was transformed into the *Agrobacterium tumefaciens* strain EHA105 using the freeze–thaw transformation method. *Arabidopsis* transformants were generated by introducing 35S:GmMT-II into the wild type using the floral dip method [34]. Three independent homozygous lines from the T3 generations were obtained using RT-PCR and immunoblotting, and these seeds were used for subsequent experiments. In addition, we obtained 3 *GmMT-II* overexpressing *Arabidopsis* lines, namely L2, L3, and the L4 in the experiment.

### 4.7. Measurement of Lipid Peroxidation (MDA)

Samples were homogenized in trichloroacetic acid (TCA, 5 %). After centrifugation at 4000 g for 10 min, 2 mL of supernatant was incorporated with thiobarbituric acid (TBA, 0.67 %) of the same volume, and the obtained mixture was centrifuged at 3000 g for 15 min after heating for 30 min using boiling water bath. The absorbance of the supernatant was measured at 450, 532, and 600 nm, which were used to calculate the MDA content according to the method of Jahan et al. [35].

### 4.8. Histochemical Analysis of Hydrogen Peroxide (H_2_O_2_) and Reactive Oxygen Species (ROS) Analysis

For the investigation of H_2_O_2_ accumulation, leaf samples were collected from transgenic *Arabidopsis* lines and wild-type *Arabidopsis* plants grown under HTH stress and staining with DAB (Diaminobenzidine) [18]. The collected leaves samples were cut into 5 mm size and incubated for 8 h in 1mg/mL DAB solution (pH = 3.8). Then, the leaves were transferred to 95% boiling ethanol for 10 min to remove the leaf pigments before the leaf segments were cleared with saturated chloral hydrate. The processed samples were mounted on a glass slide with 1:1 (glycerol and water) and observed under a microscope (Leica model DM5000B; Leica Microsystems GmbH, Wetzlar, Germany). Similarly, ROS production was quantified using a luminol-based assay [36].

### 4.9. Determination of H_2_O_2_ Content

The H_2_O_2_ content was measured according to the method developed by Barja G. [37]. After grinding the samples in TCA (0.1 %), the homogenates were centrifuged at 12,000 g for 20 min. A total of 0.2 mL supernatant, 1 mL of 1 M KI solution and 0.25 mL of 0.1 M potassium phosphate buffer (pH 7.8) were mixed together and placed in darkness for 1 h. The absorbance of the mixture was read at 390 nm, and the H_2_O_2_ concentration was calculated with a standard curve based on serial concentration gradient of H_2_O_2_ and the corresponding absorbance.

### 4.10. Determination of ROS Production

The samples were floated in an incubation buffer (10 mMHepes-NaOH, 0.2 mM CaCl_2_, pH 5.7) at 25 °C for 30 min, avoiding light, with adding 10 mM H_2_DCFDA, and then washed 3 times in the incubation buffer. The seeds were observed for fluorescence using a confocal microscope (Zeiss, Germany), with excitation at 488 nm and emission at 525 nm.

### 4.11. Analysis of Antioxidants Enzyme Activity

For antioxidant enzyme activities analysis, the samples were prepared according to Kim et al. [38]. Briefly, frozen composite leaf samples were homogenized in phosphate buffer (pH 7.4) according to the instructions of the company followed by centrifugation for 15 min at 12,000 g. After centrifugation, the supernatant was transferred to a new tube to determine the enzyme essays. The superoxide dismutase (SOD), catalase (CAT), and peroxidase (POD) content was determined according to the Kits description (Jiancheng Bioengineering Institute, Nanjing, China).

### 4.12. Statistical Analysis

All experimental data were statistically analyzed with three biological replications, and the values are expressed as the mean value with standard deviation (SD ±) and the results were statistically analyzed using one-way ANOVA (analysis of variance) through SPSS 22.0 software (SPSS Inc., Chicago, IL, USA) and the significant differences among the treatment were evaluated using Tukey’s test (*p* < 0.01). The graphs were made by Graph Pad Prism 5 (GraphPad software, San Diego, CA, USA, accessed on 27 March 2022). 

## 5. Conclusions

In summary, the higher seed germination rate of transgenic *Arabidopsis* lines compared to wild-type *Arabidopsis* plants under HTH stress conditions indicates the importance of *GmMT-II* in plant stress tolerance. Furthermore, a lower accumulation of H_2_O_2_ content and ROS molecules and higher antioxidant activity increased HTH tolerance. These results suggest that the role of GmMT-II influences plant stress tolerance to high temperature and humidity by inhibiting the accumulation of H_2_O_2_ and ROS. 

## Figures and Tables

**Figure 1 plants-11-01503-f001:**
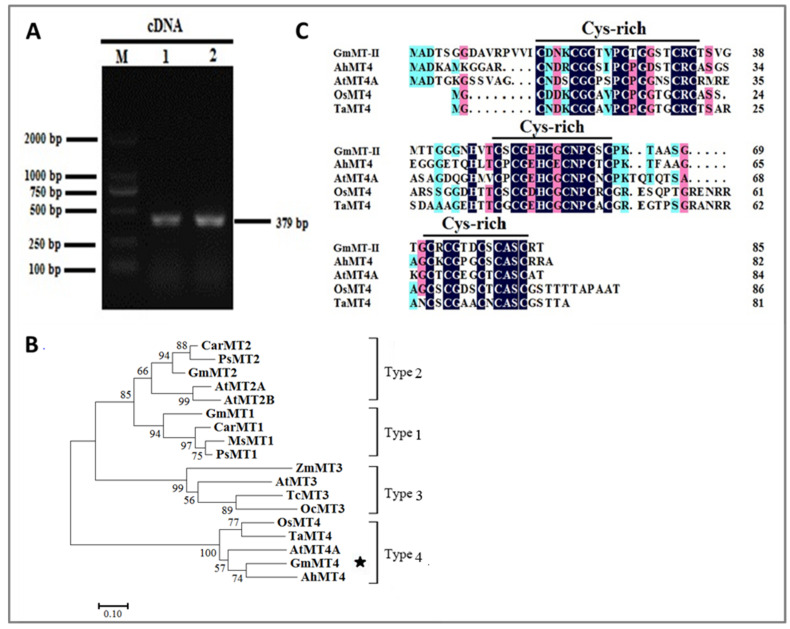
Isolation and identification of GmMT-II from Xiangdou No. 3 and Ningzhen No. 1 soybean cultivars (**A**). Isolation of cDNA of *GmMT-II* gene from soybean cultivars; (**B**) Phylogenetic tree analysis was performed by the MEGA 6 program with the neighbor joining method and with 1000 replicates. The phylogenetic tree was constructed based on the aligned amino acid sequence of 18 homologous metallothioneins (MTs) proteins. The asterisks indicate GmMT-II. (**C**) Alignment of amino acid sequences of GmMT-II in soybean cultivars with MT-II-like proteins from different plants (Ah—*Arachis hypogaea*; At—*Arabidopsis thaliana*; Os—*Oryza sativa*; and Ta—*Triticum aestivum*).

**Figure 2 plants-11-01503-f002:**
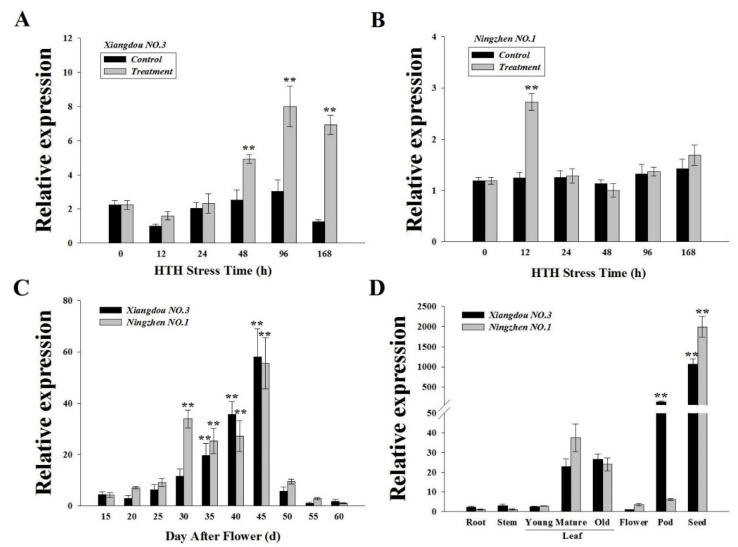
(**A**–**B**). Transcript level of *GmMT-II* in Xiangdou No. 3 and Ningzhen No. 1 soybean cultivars in different hour intervals under HTH stress conditions. (**C**). Expression level of *GmMT-II* in Xiangdou No. 3 and Ningzhen No. 1 soybean cultivars quantified days after flowering. (**D**). Expression level of *GmMT-II* in different tissues of Xiangdou No. 3 and Ningzhen No. 1 soybean cultivars. Results are shown as mean ± SD for three independent replicates. ** denotes significance at *p* < 0.01 level, according to Tukey’s correction (*p* < 0.01). Scale bars: 1 mm.

**Figure 3 plants-11-01503-f003:**
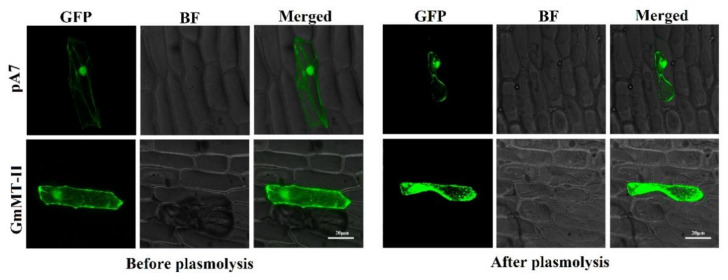
Subcellular localization of GmMT-II in onion epidermis. Confocal images showing the localization of GmMT-II in soybean mesophyll protoplasts and the empty vector (pA7) was used as negative control. The GFP signal was detected using confocal microscopy (GFP, green), bright field (gray), and merged signals are shown. Scale bars—20 µm.

**Figure 4 plants-11-01503-f004:**
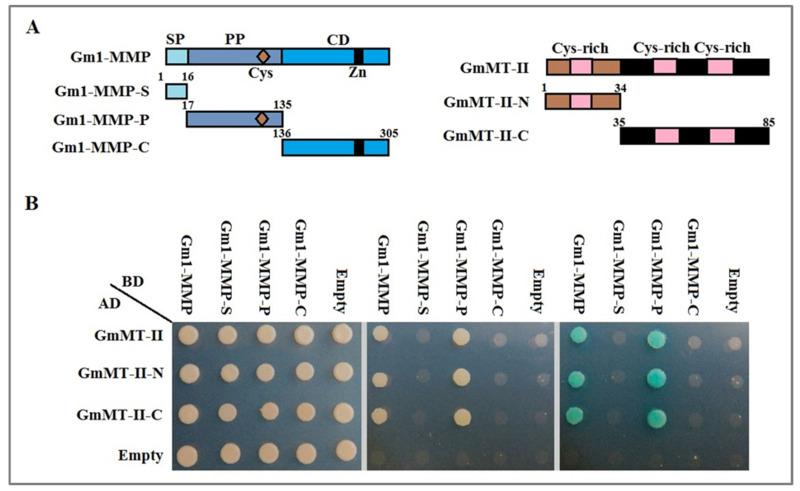
Screening of Gm1-MMP and GmMT-II interaction sites in Yeast. (**A**) Binding site prediction. (**B**) Different binding sites point to point verification.

**Figure 5 plants-11-01503-f005:**
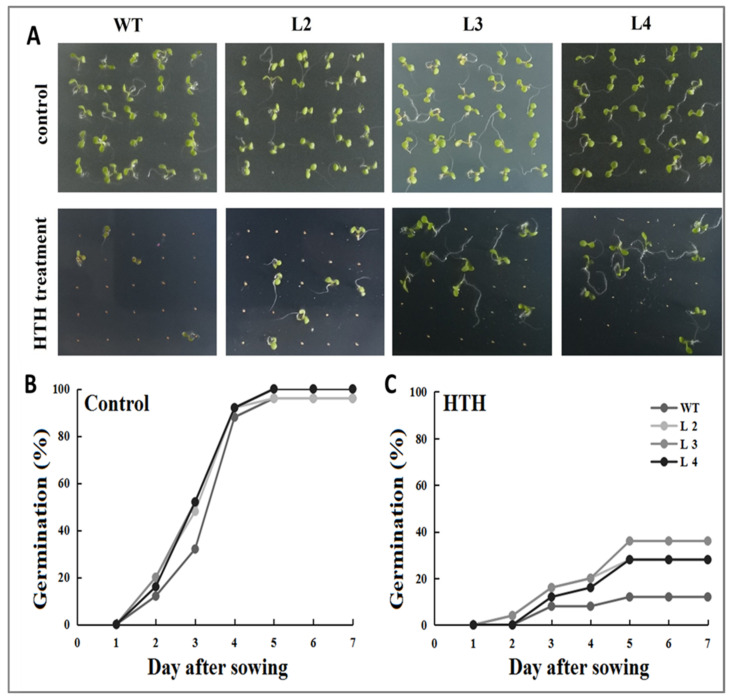
Evaluation of overexpression transgenic *Arabidopsis* lines and wild-type *Arabidopsis* seeds germination under HTH stress conditions. (**A**) Seed germination of overexpression transgenic *Arabidopsis* lines and wild-type *Arabidopsis* on growing medium under control and HTH stress conditions. (**B**–**C**) Germination rates of overexpression transgenic *Arabidopsis* lines and wild-type *Arabidopsis* seeds under control and HTH stress conditions. Each data point is the average of three experimental repetitions.

**Figure 6 plants-11-01503-f006:**
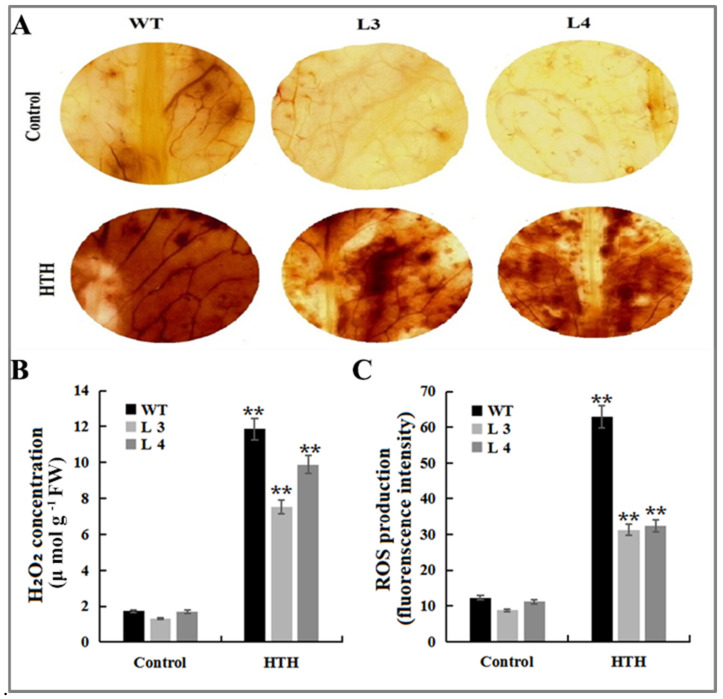
Interactive effects of high temperature and high humidity on (**A**) accumulation of hydrogen peroxide, (**B**) hydrogen peroxide content, (**C**) accumulation of superoxide anion. ** denotes significance at *p* < 0.01 level, according to Tukey’s correction (*p* < 0.01).

**Figure 7 plants-11-01503-f007:**
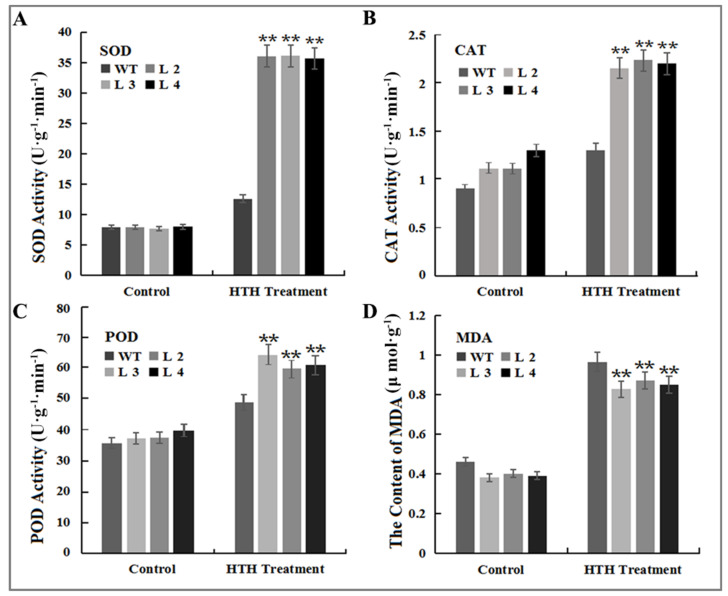
Antioxidant enzyme activities and MDA content in leaves of wild-type (WT) and overexpression transgenic *Arabidopsis* lines (L3, L4, and LS5) under HTH stress conditions. (**A**) Superoxide dismutase (SOD); (**B**) Catalase (CAT); (**C**) Peroxidase (POD); (**D)**. Malondialdehyde (MDA). ** denotes significance at *p* < 0.01 level, according to Tukey’s correction (*p* < 0.01); vertical bars indicate standard errors of each mean value (n = 3).

## Data Availability

The data presented in this study are available on request from the corresponding author.

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
