# Peer review of "Isolation and Characterization of the GmMT-II Gene and Its Role in Response to High Temperature and Humidity Stress in Glycine max"

_plants, 2022, doi:10.3390/plants11111503_

Round 1
Reviewer 1 Report
The research article "Isolation and characterization of the GmMT-II gene and its response to high temperature and humidity stress in Glycine max" by Sushuang Liu et al., describes sequence analysis, expression, subcellular localization, protein-protein interaction of GmMT-II and response of GmMT-II transgenic plants to high temperature and humidity stress (HTS).
The abstract and the article has numerous English language and formatting issues which need to be corrected by either authors or language editing services. For instance:
line 16: “Metallothioneins (MTs) are polypeptide-encoded gene” → polypeptide-encod_ing_ gene
line 19: “However, the function of MTs in higher plants is still largely elucidated.” → largely _not_ eludicated or, better, is still largely unknown
line 22: “The subcellular location of the GmMT-II-GFP fusion protein was clearly located” → The GmMT-II-GFP fusion protein was clearly located in
…
line 240-241: “In consistent” → inconsistent
line 5,7: The name of the Academy is formatted in superscript
line 320: “Suaeda salsa” should be italicized
line 327, 337: inconsistent italization
line 202: the figure 4 is on line 202, while it’s description is at line 208
and so on.
With the exception of problems with English language and formatting, the introduction fully covers the current state of knowledge and shows why the undertaken research was performed. Materials and methods are written a bit too brief and need some minor additions. Results are sound and convincing except for the omission of the description of LS2-4 mutants and the vector used.
I have some remarks that should be addressed:
1. Line 180 is stating that the gene gun method of onion epidermal cells transfection was used, whereas line 403 states “transient transformation of onion epidermal was performed through infiltration with syinge” which statement is correct?
2. Please provide a map of pA7-GFP vector or a citation to the corresponding article, describing this vector.
3. Figure 3. Scale bars are mentioned in figure description but are absent on the micro-photographs. Please add them.
4. At line 212 three transgenic Arabidopsis lines (LS2, LS3, and the LS4) are mentioned for the first time, but the description of these lines is absent from the article. Please provide the description of these lines: are they acquired from some collection or produced by the authors; what are the difference between these lines – are they just the biological replicates from different transformation experiments described at line 387 in materials and methods section? Please clarify this in the article after first mentioning these lines.
5. Figure 6 is missing the “D” section of the panel mentioned in the description as “(D) Superoxide anion production rate in tomato seedlings.”. Moreover in the article the lines are denoted as LS2, LS3, and the LS4, while in this figure the abbreviations L3, L4 and L-3, L-4 are used. Are the L3,L4,L-3,L-4 the same as LS3, LS4? Is this figure provides the data on Arabidopsis mutants or tomato seedlings?
6. Thorough the article the combined stress of high temperature and humidity is researched, but figure 6 description introduces “Interactive effects of Putrescine and high temperature”. Putrescine stress is not mentioned anywhere else in the article. What kind of stress was actually used for figure 6?
7. At line 409 simultaneous excitation of 488 and 568 lines with the detection of 500-530 nm is used. Why such a strange combination of excitation wavelengths was used? Was some kind of photoswitcheable GFP used or some other label that is excited by 568nm line? Please explain the need of 568 nm line in the acquisition.
8. During the analysis of GmMT-II gene CDSs in Xiangdou No.3 and Ningzhen No.1 soybean cultivars, were any differences in nucleotide sequence found or these sequences were absolutely identical? If so, since a clear difference in GmMT-II gene expression was found, were any attempts made to sequence the UTRs or Cis-regulatory elements to explain such a difference?
I think this article is of interest to the readers of MDPI Plants and can be published after a major revision concerning aforementioned scientific remarks and multiple issues with English language and formatting.
Author Response
- 第 180 行是用基因枪法转染洋葱表皮细胞,而第 403 行是“通过 syinge 浸润进行洋葱表皮的瞬时转化”,哪种说法是正确的?
反应:通过粒子轰击法将重组质粒pA7-GmMT-II和空载体pA7-GFP导入洋葱表皮细胞和烟叶细胞。我们在文章中进行了修改。
- 请提供 pA7-GFP 载体图或对相应文章的引用,描述该载体。
回应:载体耐药性是氨苄青霉素。已提供 pA7-GFP 载体图谱。
- 图 3. 比例尺在图描述中提到,但在显微照片中没有。请添加它们。
回应:图 3 中添加了比例尺。
- At line 212 three transgenic Arabidopsis lines (LS2, LS3, and the LS4) are mentioned for the first time, but the description of these lines is absent from the article. Please provide the description of these lines: are they acquired from some collection or produced by the authors; what are the difference between these lines – are they just the biological replicates from different transformation experiments described at line 387 in materials and methods section? Please clarify this in the article after first mentioning these lines.
Response: we have added the descriptions “In addition, we obtained 3 GmMT-II overexpressing Arabidopsis lines, namely L2, L3, and the L4 in the experiment” in the materials and methods. In addition, they are just the biological replicates from different transformation experiments produced by the authors.
- Figure 6 is missing the “D” section of the panel mentioned in the description as “(D) Superoxide anion production rate in tomato seedlings”. Moreover in the article the lines are denoted as LS2, LS3, and the LS4, while in this figure the abbreviations L3, L4 and L-3, L-4 are used. Are the L3,L4,L-3,L-4 the same as LS3, LS4? Is this figure provides the data on Arabidopsis mutants or tomato seedlings?
Response: We have deleted the “(D) Superoxide anion production rate in tomato seedlings. The data denote the mean value ± standard error (n= 3)” in the article. This figure provides the data was on Arabidopsis mutants.
The L3,L4,L-3,L-4 the same as LS3, LS4. In addition, we have unified the full text into L2, L3, L4.
- Thorough the article the combined stress of high temperature and humidity is researched, but figure 6 description introduces “Interactive effects of Putrescine and high temperature”. Putrescine stress is not mentioned anywhere else in the article. What kind of stress was actually used for figure 6?
Response: The high temperature and high humidity was used for figure 6. We have deleted the Putrescine stress in the article.
- At line 409 simultaneous excitation of 488 and 568 lines with the detection of 500-530 nm is used. Why such a strange combination of excitation wavelengths was used? Was some kind of photoswitcheable GFP used or some other label that is excited by 568nm line? Please explain the need of 568 nm line in the acquisition.
Response: The excitation wavelength of GFP is 488nm and the emission wavelength is 507nm. We have deleted the mistake in the article.
- During the analysis of GmMT-II gene CDSs in Xiangdou No.3 and Ningzhen No.1 soybean cultivars, were any differences in nucleotide sequence found or these sequences were absolutely identical? If so, since a clear difference in GmMT-II gene expression was found, were any attempts made to sequence the UTRs or Cis-regulatory elements to explain such a difference?
Response: The GmMT-II gene CDSs in Xiangdou No.3 and Ningzhen No.1 soybean were absolutely identical. We found that there are differences in the expression of GmMT-II in the two varieties by analyzing the transcription level of GmMT-II in different tissues, seed development process and under high temperature and high humidity stress. In order to study the reason, we isolated the promoter sequences of the GmMT-II gene in the two varieties which were found to be consistent. So we speculate that the expression of the gene at the transcription level is very complex and may be affected by many pathways. In addition, this gene may also regulate many personality, namely has the function of one cause and multiple effects. In multiple studies of our research group, we found that the CDS sequences of different genes are consistent, but there are differences in the expression between the two varieties. Our lab have published many articles about this, such as Zhou YL, Wang S, Hu HM, Shen YZ, Zhu YJ, Liu XL, Wei JP, Yu XX, Liu SS, Ma H. GmCDPKSK5 Interacting with GmFAD2-1B Participates in Regulation of Seed Development in Soybean Under High Temperature and Humidity Stress. Plant Molecular Biology Reporter (2022).
Wang S, Tao Y, Zhou YL, Niu J, Shu YJ, Yu XW, Liu SS, Chen M, Gu WH, Ma H. Translationally controlled tumor protein GmTCTP interacts with GmCDPKSK5 in response to high temperature and humidity stress during soybean seed development. Plant Growth Regulation, 2017, 82: 187-200.
Liu SS, Jia YH, Zhu YJ, Zhou YL, Shen YZ, Wei JP, Liu XL, Liu YM, Gu WH, Ma H. Soybean matrix metalloproteinase Gm2-MMP, relates to growth and development, and confers enhanced tolerance to high temperature and humidity stress in transgenic Arabidopsis. Plant Molecular Biology Reporter, 2018, 36(1):94-106.
Shu YJ, Tao Y, Wang S, Huang LY, Yu XW, Wang ZK, Chen M, Gu WH, Ma H. GmSBH1, a homeobox transcription factor gene, relates to growth and development and involves in response to high temperature and humidity stress in soybean. Plant Cell Reports, 2015, 34: 1927-1937.
此外,虽然两个品种的 GmMT-II 存在一些差异,但 GmMT-II 在种子中的表达水平很高,表明它参与了种子的发育,以及对高温高湿度压力。我们更关注压力处理和控制之间的差异。结果表明,GmMT-II在胁迫处理和对照之间存在显着差异,表明它参与了对高温高湿胁迫的响应。至于GmMT-II基因为何存在于不同品种之间的转录水平,我们将在未来进一步研究,如从DNA甲基化等方面进一步研究。

Reviewer 2 Report
This is a useful study about GmMT-II gene in Glycine max. Results are interesting and correlate with the data about metallothioneins.
In introduction (line 97) there is no reference for study of GmSBH1.
Line 155 – it seems it should be Figure 2 instead of 1
Check the whole part 2.2 – everywhere only Figure 2 B,D is mentioned.
Add more references for Figure 4 in part 2.4. The same is for part 2.5. In addition, it is good to explain the kind of transgenic plants of Arabidopsis: overexpressed or silenced. In Materials and methods there is an explanation, but in my opinion, it would be good to repeat it here.
Chapter 2.6 begins with experiments about MDA and figure 7 and then the text is about results on Figure 6. The talk about Figure7 cannot be before figure 6. Line 234 – this is the first time MDA is mentioned. Not everybody knows what is this. So give a short explanation of MDA.
Figures 6 and 7 contain different images A, B, C, D and in the text there are references as figure 6 or 7. This is not comfortable for a reader. Ad references for every image. For example, lines 265-268 are about SOD, so Figure 7 A should be mentioned.
Line 272 – Figure 5C is a mistake.
What is the species name of Arabidopsis plants that were used? Was it cultivar or wild type line?
MS medium is well known, but it is good to give references.
It is not clear about the control conditions of growing of soybean plants. Lines 363-364 –the conditions of HTH are repeated.
Line 390 – give the number of the Table.
In discussion, it would be good to give your thoughts and suggestions about the differences in relative expression of GmMT-II in Xiangdou No.3 and Ningzhen No.1 soybean 157 cultivars.
In my file there is no Table S1.
Author Response
在介绍(第 97 行)中,没有研究 GmSBH1 的参考资料。
回应:我们有研究GmSBH1的正确参考。
第 155 行 – 似乎应该是图 2 而不是图 1
回应:我们已经修改了文章中的错误。
检查整个第 2.2 部分——在所有地方都只提到了图 2 B、D。
回应:之前,图 2 A 和 C 被误写为图 1 A 和 C,现已修改。
在第 2.4 部分中为图 4 添加更多参考。第 2.5 部分也是如此。此外,最好解释一下拟南芥的转基因植物种类:过表达或沉默。在材料和方法中有一个解释,但在我看来,在这里重复一下会很好。
Response: the references have been added and the kind of transgenic plants of Arabidopsis have been explained.
Chapter 2.6 begins with experiments about MDA and figure 7 and then the text is about results on Figure 6. The talk about Figure7 cannot be before figure 6. Line 234 – this is the first time MDA is mentioned. Not everybody knows what is this. So give a short explanation of MDA.
Response: We have adjusted the content of Figure 7 in the article, ranking after figure 6. The full name of MDA, malondialdehyde, has been added where it first appears.
Figures 6 and 7 contain different images A, B, C, D and in the text there are references as figure 6 or 7. This is not comfortable for a reader. Add references for every image. For example, lines 265-268 are about SOD, so Figure 7 A should be mentioned.
Response: We have added the Figure A B C and D in the correct place, individually.
Line 272 – Figure 5C is a mistake.
Response: we have revised the Figure 5C to Figure 7C.
What is the species name of Arabidopsis plants that were used? Was it cultivar or wild type line?
Response: Wild type Arabidopsis plants were used.
MS medium is well known, but it is good to give references.
Response: We have added the reference “Kawka B, Kwiecień I, Ekiert H. Influence of Culture Medium Composition and Light Conditions on the Accumulation of Bioactive Compounds in Shoot Cultures of Scutellaria lateriflora L. (American Skullcap) Grown In Vitro. Appl Biochem Biotechnol. 2017 Dec;183(4):1414-1425. doi: 10.1007/s12010-017-2508-2. Epub 2017 Jun 1. PMID: 28573603; PMCID: PMC5698381” where it first appears.
It is not clear about the control conditions of growing of soybean plants. Lines 363-364 –the conditions of HTH are repeated.
Response: Soybean seeds were sown in plastic pots and managed under the normal condition [30℃/20℃, 70% RH, and 10 h/14 h (light/dark)]. HTH stress on potted soybean plants (R 7 stage) was performed as previously described(Wang et al., 2012). Treated plants were transferred to growth cabinet at 40℃/30℃, 100%/70% humidity (RH) and 10 h/14 h (light/dark) light cycle for 7 d. Control plants with the same developmental progression were cultivated under the normal condition. We have adjusted the content in the materials and methods.
Line 390 – give the number of the Table.
Response: we have added the Table S1.
In discussion, it would be good to give your thoughts and suggestions about the differences in relative expression of GmMT-II in Xiangdou No.3 and Ningzhen No.1 soybean 157 cultivars.
Response: We have provided our thoughts and suggestions about the differences in relative expression of GmMT-II in Xiangdou No.3 and Ningzhen No.1 soybean 157 cultivars in discussion as follows.
The GmMT-II gene CDSs in Xiangdou No.3 and Ningzhen No.1 soybean were absolutely identical. We found that there are differences in the expression of GmMT-II in the two varieties by analyzing the transcription level of GmMT-II in different tissues, seed development process and under high temperature and high humidity stress. In order to study the reason, we isolated the promoter sequences of the GmMT-II gene in the two varieties which were found to be consistent. So we speculate that the expression of the gene at the transcription level is very complex and may be affected by many pathways. In addition, this gene may also regulate many personality, namely has the function of one cause and multiple effects. In multiple studies of our research group, we found that the CDS sequences of different genes are consistent, but there are differences in the expression between the two varieties.
In addition, although there are some differences in GmMT-II in the two varieties, the expression level of GmMT-II in seeds is very high, indicating that it is involved in the development of seeds, and for the response to high temperature and high humidity stress. We pay more attention to the difference between the stress treatment and the control. The results showed that GmMT-II was significantly different between the stress treatment and the control, indicating that it was involved in the response to high temperature and high humidity stress. As for why the GmMT-II gene exists at the transcription level between different varieties, we will further study in the future, such as further study from DNA methylation and so on.
In my file there is no Table S1
Response: we have provided the Table S1 as follows.
Table S1 The primers used in allexperiments
Primer names |
Primer sequence (5’-3’) |
Experiments |
ORF-GmMT-II-F |
AGAGAGAAAAATGGCTGATACAAGT |
segregation of genes |
ORF-GmMT-II-R |
AAATCACATCCACAATTTGAACA |
|
pBT3Gm1-MMP-F |
TTGGCCATTACGGCCATGACTCTCCGCAACCACC |
Yeast Two-Hybrid Rotation Validation |
pBT3Gm1-MMP-R |
AAGGCCGAGGCGGCCCCGGGGTTGATACCATAGAGCTTTCG |
|
pPR3N-GmMT-F |
TTGGCCATTACGGCCATGGCTGATACAAGTGGAG |
|
pPR3N-GmMT-R |
AAGGCCGAGGCGGCCCAGTGCGGCAAGAGGCACATGAGCA |
|
nE-Gm1-MMP-F |
GCTTCGAATTCTGCAGTCGACATGACTCTCCGCAACCACC |
BiFC experiments |
nE-Gm1-MMP-R |
GACTCTAGATCAGGTGGATCCGGGGTTGATACCATAGAGC |
|
cE-GmMT-F |
GCTTCGAATTCTGCAGTCGACATGGCTGATACAAGTGGAG |
|
cE-GmMT-R |
GACTCTAGATCAGGTGGATCCAGTGCGGCAAGAGGCACAT |
|
nEYFP-F |
GAAGAACGGCATCAAGGT |
|
nEYFP-R |
CGACAGGTTTCCCGACTG |
|
cEYFP-F |
CACTACCAGCAGAACACCC |
|
cEYFP-R |
CAGCTATGACCATGATTAC |
|
Gm1-MMP-S-F |
TTGGCCATTACGGCCATGACTCTCCGCAACCACC |
Interaction site research |
Gm1-MMP-S-R |
AAGGCCGAGGCGGCCCCTAGAATTGCAAGAGCAACCAAGAG |
|
Gm1-MMP-P-F |
TTGGCCATTACGGCCTATTTTCTTGCCACCTCAC |
|
Gm1-MMP-P-R |
AAGGCCGAGGCGGCCCCGGTATAGTCCGAGATCATGCCAAA |
|
Gm1-MMP-CF |
TTGGCCATTACGGCCTTCTTCAAAGACATGCCGC |
|
Gm1-MMP-CR |
AAGGCCGAGGCGGCCCCGGGGTTGATACCATAGAGCTTTCG |
|
GMT-NF |
TTGGCCATTACGGCCATGGCTGATACAAGTGGAG |
|
GMT-NR |
AAGGCCGAGGCGGCCCGCACCTGCAAGTGGAACCACCAGT |
|
GMT-CF |
TTGGCCATTACGGCCACAAGTGTCGGCATGACAA |
|
GMT-CR |
AAGGCCGAGGCGGCCCAGTGCGGCAAGAGGCACATGAGCA |
|
qRT-GmMT-II-F |
GCAGTGAGACCGGTGGTAATA |
逆转录定量PCR |
qRT-GmMT-II-R |
AGTTCCAGAAGCCGCAGTC |
|
肌动蛋白-F |
CCTCAACCCAAAGGTCAACAG |
|
肌动蛋白-R |
GACCAGCGAGATCCAAACGAA |
|
pA7-GmMT-II-F |
CACCATCACCATCACGCCATGATGGCTGATACAAGTGGAG |
亚细胞定位 |
pA7-GmMT-II-R |
CACTAGTACGTCGACCATGGCAGTGCGGCAAGAGGCACAT |
|
pA7-F |
AAGCAATCAAGCATTCTACTTCTAT |
|
pA7-R |
GGTAGCGGCTGGAGCACTG |

Reviewer 3 Report
Dear Authors
Undoubtedly, characterization of genes associated with abiotic stresses is one of the interesting and important studies. However, the manuscript has major problems in writing specially organization of the different parts of the manuscript. It is kind of solving a puzzle putting different part of manuscript together to understand the whole story. For example:
1-The topic should be change, Isolation and characterization of the GmMT-II gene and its response OR IT’S ROLE IN RESPONSE to high temperature and humidity stress in Glycine max SEED?
2- Abstract needs to be re-write carefully. For example, it starts with MTs gene then goes to HTH stress in soybean seeds and then a sentence ‘However, the function of MTs in higher plants is still largely elucidated’. Or in Line 24, you talk about the gene expression in two soybean cultivars but there is no information mentioned about that previously.
3- Another issue is that the result start with identification about MTs from NCBI and then the transcript level of GmMT-II BUT THE MATERIAL METHODS START WITH TRANSGENIC ARABIDOPSIS. The manuscript needs to be organize in better way.
4-Most importantly there is no information about the stress condition, plant (biological) replicates and experimental replicates for gene expression in soybean. At what stage of plant’s life cycle did you start stress and what was the temperature and % of humidity? and why? And what was the difference between night and daytime in case of temperature and lighting? Where were soybean plants grown? under field condition or in the greenhouse? What was the irrigation condition?
Best wishes
Author Response
1-主题应该是 GmMT-II 基因的变化、分离和表征及其对 Glycine max SEED 中高温和高湿应激反应的反应或作用?
回应:我们修改了标题“GmMT-II 基因的分离和表征及其在大豆中对高温和高湿胁迫的反应中的作用”。
2-摘要需要仔细重写。例如,它从 MTs 基因开始,然后进入大豆种子中的 HTH 胁迫,然后是一句话“然而,MTs 在高等植物中的功能仍然很大程度上被阐明”。或者在第 24 行,您谈到了两个大豆品种中的基因表达,但之前没有提到过相关信息。
回应:我们重新编写了这样的摘要:
High temperature and humidity stress (HTH) reduced seed development and maturity of the field-grown soybean leads to seed pre-harvest deterioration. Metallothioneins (MTs) are polypeptide-encoded genes and involved in plant growth, development, seed formation, and diverse stress response. However, the function of MTs in higher plants is still largely unknown. The highest expression of the GmMT-II gene was observed in seeds both of the soybean cultivars, as compared to other plant tissues. Similarly, gene expression was higher 45 days after flowering followed by 30, 40 and 35 days. Furthermore, the GmMT-II transcript levels were significantly higher at 96 and 12 h in the cultivars Xiangdou No.3 and Ningzhen No.1 under HTH stress, respectively. Herein, we isolated and characterized soybean metallothionein II gene. The full-length fragment contains 255 bp and is encodes 85 amino acids and located in the HD domain and the N-terminal non-conservative region. The GmMT-II-GFP fusion protein was clearly located in the nucleus, cytoplasm, and cell membrane. In addition, it was found that when the Gm1-MMP protein was deleted, the propeptide region of the Gm1MMP protein could bind to the GmMT-II but not the signal peptide region or the catalytic region. GmMT-II overexpression in transgenic Arabidopsis increased seed germination and germination rate under HTH conditions, conferring enhanced resistance to HTH stress. GmMT-II overexpressing plants suffered less oxidative damage under HTH stress, as reflected by lower MDA and H2O2 content and ROS production than WT plants. In addition, the antioxidant enzymes namely SOD, CAT, and POD activity was significantly higher in all transgenic Arabidopsis lines under HTH stress, while lowered activity was noticed in WT plants. Our results suggested that GmMT-II is related to growth and development and confers enhanced HTH stress tolerance in plants by reduction of oxidative molecules through activation of antioxidant activities. These findings will be helpful for us in further understanding of the biological functions of MT-II in plant.
3- Another issue is that the result start with identification about MTs from NCBI and then the transcript level of GmMT-II BUT THE MATERIAL METHODS START WITH TRANSGENIC ARABIDOPSIS. The manuscript needs to be organized in better way.
Response: We have swapped positions of the original 4.1 and 4.2.
4-Most importantly there is no information about the stress condition, plant (biological) replicates and experimental replicates for gene expression in soybean. At what stage of plant’s life cycle did you start stress and what was the temperature and % of humidity? and why? And what was the difference between night and daytime in case of temperature and lighting? Where were soybean plants grown? under field condition or in the greenhouse? What was the irrigation condition?
Response: Soybean cv. Ningzhen No. 1,a pre-harvest seeddeterioration sensitive cultivar, and cv. Xiangdou No. 3, a pre-harvest seed deterioration resistant cultivar, were used in this study (Wang et al. 2012). Seeds were sown in plastic pots and normally managed [30℃/20℃, 70% RH, and 10 h/14 h (light/dark)]. The organs including young leaves, stems, and roots at V1 stage; flowers and mature leaves at R2 stage; pods and old leavesatR4 stage;and matureseeds at R8 stage were used for the tissue specific expression assay. HTH stress on potted soybean plants (R 7 period) was performed according to Wang et al. (2012). Plants used for treatment [40℃/30℃, 100%/70% humidity (RH) and 10 h/14 h (light/dark) light cycle] were transferred to growth cabinet for 7 days. Meanwhile, the control plants continued to grow under normal condition. Seeds were collected from the central part of 10 control plants and 10 treated plants after stress 24, 96, and 168 h, each sample contains three independent biological replicates. All collected samples were refrigerated in liquid nitrogen and kept subsequently at an ultra-low temperature refrigerator for total RNA extraction.

Round 2
Reviewer 1 Report
The article was already significantly improved by the authors. I think that only some minor corrections are still needed before accepting the article for publication.
Some minor problems which need correction:
line 16: “Metallothioneins (MTs) are polypeptide-encoded gene” → polypeptide-encod_ing_ gene. (Due to the fact that the gene is coding for amino-acid sequence of the peptide, and not vice-versa)
Inconsistent naming of the line L4:
Line382 “Both the overexpression transgenic Arabidopsis lines (L2, L3, and L4) and ...”
Line 212, 219, 243 249, 267, 275, 284 – The line name is given as "L4L4" instead of "L4" used in figures and materials and methods section
Lines 322-323 hypenation is missing “abso <new line> lutely” -> “abso- <new line> lutely”
Line 329 I think it would be better to use "traits" instead of "personality"
Line 416 Table number and table contents, mentioned on this line is missing
line 436-437 “The GFP channel was acquired by simultaneous scanning using 488 nm laser lines for excitation.” → The GFP channel was acquired by excitation at 488nm with detection at 500-530 nm.
Author Response
Thank you very much for your valuable comments, we have revised the text one by one according to your comments.

Reviewer 3 Report
Dear Authors
Thank you so much for re-submitting the manuscript. However, many of the main problems are still remining regarding to the presenting and writing style and language. It seems that manuscript was written by someone who has no experience in manuscript preparation and submission. In my previous comments, I strongly suggested to re-write the abstract part but still the main problems remining. For instance, in line 20 & 21 you again mention ‘The highest expression of the GmMT-II gene was observed in seeds both of the soybean cultivars’. Which both? Should readers guess that? Or everyone knows which cultivars you work with? You might write in 2 sensitive and resistant soybean cultivars or should mention the name of them or in the worst case you should write in 2 different soybean cultivars. I have specially mentioned about this part in my previous comments, and you haven’t changed that. Moreover, the grammar of whole sentences is wrong.
My previous comments were just some examples, and they were not the only problems. I made those comments as examples to show authors what they need to be careful about, when they are writing a manuscript. Here in new version of your manuscript you mentioned about the stress conditions but there are still many problems. For example, in the line 375 in the method parts ‘The GmMT-II homologous sequences and amino acid sequences of other plant species were retrieved from the NCBI database (https://www.ncbi.nlm.nih.gov/) for sequence comparison’. OTHER PLANT SPECIES? How many plant species? And which species exactly?????? Should readers guess that too? There is also no information in the results part. In line 111 ‘GmMT-II CDS sequences as the template’, that sequence is from which plant species? Soybean? Or maybe Arabidopsis? It is not clear at al.
Regards
Author Response
Thank you so much for re-submitting the manuscript. However, many of the main problems are still remining regarding to the presenting and writing style and language. It seems that manuscript was written by someone who has no experience in manuscript preparation and submission. In my previous comments, I strongly suggested to re-write the abstract part but still the main problems remining. For instance, in line 20 & 21 you again mention ‘The highest expression of the GmMT-II gene was observed in seeds both of the soybean cultivars’. Which both? Should readers guess that? Or everyone knows which cultivars you work with? You might write in 2 sensitive and resistant soybean cultivars or should mention the name of them or in the worst case you should write in 2 different soybean cultivars. I have specially mentioned about this part in my previous comments, and you haven’t changed that. Moreover, the grammar of whole sentences is wrong.
Response: Thank you very much for your valuable comments, we have revised into ‘The highest expression of the GmMT-II gene was observed in seeds both of the soybean Xiangdou No.3 and Ningzhen No.1 cultivars, as compared to root, stem, leaf and flower etc. tissues’. In addition, we have checked the grammar. In addition, we have revised the abstract carefully.
My previous comments were just some examples, and they were not the only problems. I made those comments as examples to show authors what they need to be careful about, when they are writing a manuscript. Here in new version of your manuscript you mentioned about the stress conditions but there are still many problems. For example, in the line 375 in the method parts ‘The GmMT-II homologous sequences and amino acid sequences of other plant species were retrieved from the NCBI database (https://www.ncbi.nlm.nih.gov/) for sequence comparison’. OTHER PLANT SPECIES? How many plant species? And which species exactly?????? Should readers guess that too? There is also no information in the results part. In line 111 ‘GmMT-II CDS sequences as the template’, that sequence is from which plant species? Soybean? Or maybe Arabidopsis? It is not clear at all.
Response: Thank you very much for your valuable comments The number of 0ther plant species are 11, and they are Ah: Arachis hypogaea, At: Arabidopsis thaliana, Car: Cicer arietinum, Gm: Glycine max, Ms: Medicago sativa, Oc: Oryza coarctata, Os: Oryza sativa, Ps: Pisum sativum, Ta: Triticum aestivum, Tc: Thlaspi caerulescens and Zm: Zea mays. In addition, we also added in the results. About the problem of line 111, the sequence is from soybean.

Round 3
Reviewer 3 Report
Dear authors,
Thanks for revision of your manuscript. However, there some small problems should be addressed. About the Abstract, I can see very nice abstract without problem in the plants website:
Abstract
Metallothioneins (MTs) are polypeptide-encoded genes and involved in plant growth, development, seed formation, and diverse stress response. High temperature and humidity stress (HTH) reduced seed development and maturity of the field-grown soybean leads to seed pre-harvest deterioration. However, the function of MTs in higher plants is still largely elucidated. Herein, we isolated and characterized soybean metallothionein II gene. The full-length fragment contains 255 bp and is encodes 85 amino acids and located in the HD domain and the N-terminal non-conservative region. The subcellular location of the GmMT-II-GFP fusion protein was clearly located in the nucleus, cytoplasm, and cell membrane. The highest expression of the GmMT-II gene was observed in seeds both of the soybean cultivars, as compared to other plant tissues. Similarly, gene expression was higher 45 days after flowering followed by 30, 40 and 35 days. Furthermore, the GmMT-II transcript levels were significantly higher at 96 and 12 h in the cultivars Xiangdou No.3 and Ningzhen No.1 under HTH stress, respectively. In addition, it was found that when the Gm1-MMP protein was deleted, the propeptide region of the Gm1MMP protein could bind to the GmMT-II but not the signal peptide region or the catalytic region. GmMT-II overexpression in transgenic Arabidopsis increased seed germination and germination rate under HTH conditions, conferring enhanced resistance to HTH stress. GmMT-II overexpressing plants suffered less oxidative damage under HTH stress, as reflected by lower MDA and H2O2 content and ROS production than WT plants. In addition, the antioxidant enzymes namely SOD, CAT, and POD activity was significantly higher in all transgenic Arabidopsis lines under HTH stress, while lowered activity was noticed in WT plants. Our results suggested that GmMT-II is related to growth and development and confers enhanced HTH stress tolerance in plants by reduction of oxidative molecules through activation of antioxidant activities. These findings will be helpful for us in further understanding of the biological functions of MT-II in plant.
This one is totally fine. But there is different abstract in the submitted manuscript. It seems you forgot to replace the edited Abstract in the main paper. Replace your abstract with this one above.
In the key word you must ad 'GmMT II' too
Line 48: ‘and these stresses alter plant physiological and biochemical changes including’ Alter already means change. Replace the word change with activities.
You have big problems in references sometimes it is ‘., ‘ for example (Hasanuzzaman et al., 2016 sometimes it is only ‘ . ‘ for example (Liu et al. 2018) and sometimes only ‘,’ for example ( Jumrani and Bhatia, 2018). This is really sad and shocking that there are 6 authors in this manuscript and none of you care about this important and very simple thing. It seems some authors even didn’t read the manuscript. It is annoying that a reviewer or editor should spend time just to let you know that how should you make citations.
Line 81: delete ‘Further ‘ in : Further, MT proteins are classified into three sub-classes:
Line 82: delete ‘of these’
Results
Line 111: You just mentioned once Figure 1 and then nothing. In line 111 should be Figure 1, A and then when you are talking about the rest of the results for example phylogenetic tree you again should show that there is a figure for that too. Would be Figure 1, C. Another important thing about Figure 1 is that the word ‘C’ is closer to the agarose gel image than being closer to the phylogenetic tree. Please change that.
What is that star in the phylogenetic tree??? If you gave there a star that means, there is something important that you need to explain. Explain that under the figure.
Line 147: You mentioned that ‘The gene expression was also higher in Ningzhen No.1 cultivar under HTH conditions and then in Line 158 you mentioned ‘Whereas, lower GmMT-II expression was observed in the HTH treatment compared to the control treatment’. Which one is correct????
You give the figure 4 in the middle of the text and then after text you give the explanation about the figure 4. How can a manuscript can be that messy after 3rd submission??????????
Line 205; parenthesis inside the parenthesis??? Why you use 2 parentheses there???? It can be Arabidopsis lines (L2, L3, L4 and WT).
Material methods:
Line 368, First mention that you get the sequence for soybean and the talk about other plants too.
Line 378, first give information about the soybean and which one is resistant or sensitive and then transgenic Arabidopsis.
Line 388, what is the V1 and R2 stages?
You still have big organization problem in the method part. After giving the information about plant material and stress then you need to give information about the gene expression same as the result part. They should be in the same order as the result part. Move the: 4.11. RNA Extraction, cDNA Library and qRT-PCR’ part exactly after the 4.2 and give more information about the kits that you used for RNA extraction and qRT-Real Time PCR and what was the thermocycler?
Best wishes,
Author Response
Thanks for revision of your manuscript. However, there some small problems should be addressed. About the Abstract, I can see very nice abstract without problem in the plants website:
Abstract
Metallothioneins (MTs) are polypeptide-encoded genes and involved in plant growth, development, seed formation, and diverse stress response. High temperature and humidity stress (HTH) reduced seed development and maturity of the field-grown soybean leads to seed pre-harvest deterioration. However, the function of MTs in higher plants is still largely elucidated. Herein, we isolated and characterized soybean metallothionein II gene. The full-length fragment contains 255 bp and is encodes 85 amino acids and located in the HD domain and the N-terminal non-conservative region. The subcellular location of the GmMT-II-GFP fusion protein was clearly located in the nucleus, cytoplasm, and cell membrane. The highest expression of the GmMT-II gene was observed in seeds both of the soybean cultivars, as compared to other plant tissues. Similarly, gene expression was higher 45 days after flowering followed by 30, 40 and 35 days. Furthermore, the GmMT-II transcript levels were significantly higher at 96 and 12 h in the cultivars Xiangdou No.3 and Ningzhen No.1 under HTH stress, respectively. In addition, it was found that when the Gm1-MMP protein was deleted, the propeptide region of the Gm1MMP protein could bind to the GmMT-II but not the signal peptide region or the catalytic region. GmMT-II overexpression in transgenic Arabidopsis increased seed germination and germination rate under HTH conditions, conferring enhanced resistance to HTH stress. GmMT-II overexpressing plants suffered less oxidative damage under HTH stress, as reflected by lower MDA and H2O2 content and ROS production than WT plants. In addition, the antioxidant enzymes namely SOD, CAT, and POD activity was significantly higher in all transgenic Arabidopsis lines under HTH stress, while lowered activity was noticed in WT plants. Our results suggested that GmMT-II is related to growth and development and confers enhanced HTH stress tolerance in plants by reduction of oxidative molecules through activation of antioxidant activities. These findings will be helpful for us in further understanding of the biological functions of MT-II in plant.
This one is totally fine. But there is different abstract in the submitted manuscript. It seems you forgot to replace the edited Abstract in the main paper. Replace your abstract with this one above.
Response: we have replaced our abstract with this one above in the paper.
In the key word you must ad 'GmMT II' too
Response: we have added the “GmMT II” in the keywords.
Line 48: ‘and these stresses alter plant physiological and biochemical changes including’ Alter already means change. Replace the word change with activities.
Response: we have replaced the word “change” with “activities”.
You have big problems in references sometimes it is ‘., ‘ for example (Hasanuzzaman et al., 2016 sometimes it is only ‘ . ‘ for example (Liu et al. 2018) and sometimes only ‘,’ for example ( Jumrani and Bhatia, 2018). This is really sad and shocking that there are 6 authors in this manuscript and none of you care about this important and very simple thing. It seems some authors even didn’t read the manuscript. It is annoying that a reviewer or editor should spend time just to let you know that how should you make citations.
Response:we are very sorry for the mistakes and we have revised in the article.
Line 81: delete ‘Further ‘ in : Further, MT proteins are classified into three sub-classes:
Response: we have deleted the ‘Further’.
Line 82: delete ‘of these’
Response: we have deleted the ‘of these’.
Results
Line 111: You just mentioned once Figure 1 and then nothing. In line 111 should be Figure 1, A and then when you are talking about the rest of the results for example phylogenetic tree you again should show that there is a figure for that too. Would be Figure 1, C. Another important thing about Figure 1 is that the word ‘C’ is closer to the agarose gel image than being closer to the phylogenetic tree. Please change that.
Response: we have revised in the article.
What is that star in the phylogenetic tree??? If you gave there a star that means, there is something important that you need to explain. Explain that under the figure.
Response: the star indicates GmMT-II, we have added the explanation under the figure 1.
Line 147: You mentioned that ‘The gene expression was also higher in Ningzhen No.1 cultivar under HTH conditions and then in Line 158 you mentioned ‘Whereas, lower GmMT-II expression was observed in the HTH treatment compared to the control treatment’. Which one is correct????
Response: we have deleted the sentence in Line 158.
You give the figure 4 in the middle of the text and then after text you give the explanation about the figure 4. How can a manuscript can be that messy after 3rd submission??????????
Response: we are very sorry for the mistakes and we have made changes in the paper.
Line 205; parenthesis inside the parenthesis??? Why you use 2 parentheses there???? It can be Arabidopsis lines (L2, L3, L4 and WT).
Response: we have revised in the paper.
Material methods:
Line 368, First mention that you get the sequence for soybean and the talk about other plants too.
Response: thanks for your suggestion and we have revised.
Line 378, first give information about the soybean and which one is resistant or sensitive and then transgenic Arabidopsis.
Response: thanks for your suggestion and we have revised.
Line 388, what is the V1 and R2 stages?
Response: the development of soybean growth stage was divided into vegetative stage and reproductive stage. The V1 refers to one trifoliolate stage and R2 refers to full bloom stage.
You still have big organization problem in the method part. After giving the information about plant material and stress then you need to give information about the gene expression same as the result part. They should be in the same order as the result part. Move the: 4.11. RNA Extraction, cDNA Library and qRT-PCR’ part exactly after the 4.2 and give more information about the kits that you used for RNA extraction and qRT-Real Time PCR and what was the thermocycler?
Response: thanks for your suggestion and we have revised.
